# Longer Survival and Preserved Liver Function after Proton Beam Therapy for Patients with Unresectable Hepatocellular Carcinoma

**Takuto Nosaka [1], Hidetaka Matsuda [1], Ryotaro Sugata [1], Yu Akazawa [1], Kazuto Takahashi [1], Tatsushi Naito [1], Masahiro Ohtani [1], Kazuyuki Kinoshita [2], Tetsuya Tsujikawa [2], Yoshitaka Sato [3], Yoshikazu Maeda [3], Hiroyasu Tamamura [3] and Yasunari Nakamoto [1,\***

[1]  Second Department of Internal Medicine, Faculty of Medical Sciences, University of Fukui, Fukui 910-1193, Japan
[2]  Department of Radiology, Faculty of Medical Sciences, University of Fukui, Fukui 910-1193, Japan
[3]  Proton Therapy Center, Fukui Prefecture Hospital, Fukui 910-8526, Japan
*   Correspondence: nakamoto-med2@med.u-fukui.ac.jp; Tel.: +81-776-61-8351

**Abstract:** Background: Proton beam therapy (PBT) has been recently reported to achieve excellent tumor control with minimal toxicity in patients with unresectable hepatocellular carcinoma (HCC). Radiofrequency ablation (RFA) combined with transcatheter arterial chemoembolization (TACE) was investigated for larger HCC. This study was designed to evaluate the therapeutic effect of PBT on unresectable HCC in comparison with TACE combined with RFA. Methods: We retrospectively analyzed 70 patients with HCC which was difficult to control by surgical resection or RFA monotherapy, 24 patients treated with PBT and 46 patients with TACE plus RFA. The therapeutic effects were assessed as local progression-free survival (PFS) and overall survival (OS). Results: The local PFS was more than 65% in 60 months for PBT and TACE plus RFA. The patients treated with PBT showed 82% OS at 60 months post-treatment. In contrast, those treated with TACE plus RFA showed 28% OS. When comparing the changes of ALBI scores in patients with different severities of chronic liver disease, the scores of PBT-treated patients were maintained at the baseline; however, those of TACE plus RFA-treated patients worsened after the treatments. Conclusions: The results indicated that PBT may show better benefits than TACE plus RFA therapy in terms of OS in patients with unresectable HCC by sparing the non-tumor liver tissues.

**Keywords:** hepatocellular carcinoma; proton beam therapy; transcatheter arterial chemoembolization; radiofrequency ablation; albumin-bilirubin score

## 1. Introduction

Hepatocellular carcinoma (HCC), the predominant form of liver cancer, is the sixth most commonly diagnosed cancer in the world and the fourth most common cause of cancer-related death [1]. Radiofrequency ablation (RFA) is the most commonly used thermal ablation modality for local treatment of HCC. Local control rates comparable to hepatectomy can be achieved with RFA alone when treating small HCCs (<2 cm) in preferred locations. However, the progression and recurrence rate of local tumors with RFA monotherapy increases sharply when larger lesions (>3 cm) are treated [2]. Even for small lesions (≤3 cm), RFA monotherapy has poor local control in the following cases: (1) the tumor is adjacent to large vessels such as the portal vein [3]; (2) the tumor is located at the subphrenic lesion [4]. To address these clinical problems, recent efforts have focused on multi-model management of HCC by combining RFA with a variety of techniques, including transcatheter arterial chemoembolization (TACE) [5,6]. TACE relies on the strong arterial supply of HCC, aiming for complete anoxia inside the malignancy, inducing an ischemic response and causing tumor necrosis. At the same time, it damages the

surrounding liver parenchyma and exacerbates the functional hepatic reserve, which leads to a worse prognosis for patients with HCC complicated with chronic liver diseases [7,8].

Proton beam therapy (PBT) has recently been applied to HCC treatment, and its local control effects and safety have been reported in various studies [9–11]. Of radiotherapy modalities, PBT has been shown to achieve excellent long-term tumor control with minimal toxicity in patients with unresectable HCC [12]. The low dose of radiation to surrounding normal tissues reduces the chances of radiation induced liver disease (RILD) [13], which is particularly important when treating patients with HCC with chronic liver diseases. Recently, the outcomes of PBT and RFA were reported in patients with recurrent/residual HCC in a phase III non-inferiority trial [14]. PBT showed 2-year local-progression-free survival values that were non-inferior to those for RFA, and it could be used safely in patients with small recurrent HCC (size < 3 cm, number ≤ 2).

In the current study, the therapeutic effect of PBT on unresectable HCC, that could not be controlled by RFA monotherapy, was compared with that of TACE plus RFA therapy. The therapeutic effects on tumor control and prognostic benefits were assessed as local progression-free survival (PFS) and overall survival (OS) after the treatments.

## 2. Materials and Methods

### 2.1. Patients

From January 2010 to February 2023, 24 patients with HCC who were treated with PBT and 46 patients who were treated with TACE followed by RFA (TACE + RFA) were included in the study. The inclusion criteria for cases treated with PBT and TACE + RFA were as follows: (1) patients who were considered to be difficult to control by resection due to comorbidities such as cardiopulmonary dysfunction, or who refused to undergo resection; (2) tumors which were adjacent to large vessels such as the portal vein or located at the subphrenic lesion; (3) ECOG performance status (PS) of 0 to 2; (4) no uncontrolled ascites; (5) no extrahepatic metastases; (6) no local recurrence from the same lesion previously treated with resection or RFA; (7) Child–Pugh class A or B liver reserve. Patient characteristics are shown in Table 1. The diagnosis of HCC was performed when the findings were typical on computed tomography (CT) or magnetic resonance imaging (MRI) according to the American Association for the Study of Liver Diseases practice guidelines for the management of HCC [15]. In cases of atypical imaging findings, a liver tumor needle biopsy was performed, and a histological diagnosis of hepatocellular carcinoma was made. This retrospective study was approved by The Research Ethics Committee of University of Fukui (20220071) and Fukui Prefecture Hospital (No. 22-21).

**Table 1.** Patients' characteristics between proton beam therapy and TACE plus RFA.

| Characteristics | PBT (*n* = 24) | TACE + RFA (*n* = 46) | *p* Value |
|---|---|---|---|
| Age, years | | | |
| Median (range) | 74 (54–88) | 74 (57–85) | 0.478 * |
| <65 | 2 (8.3) | 3 (6.5) | >0.999 † |
| ≥65 | 22 (91.7) | 43 (93.5) | |
| Gender | | | |
| Male | 16 (66.7) | 33 (71.7) | 0.785 † |
| Female | 8 (33.3) | 13 (28.3) | |
| ECOG performance status | | | |
| 0 | 21 (87.5) | 37 (80.4) | 0.526 † |
| 1 | 3 (12.5) | 9 (19.6) | |
| Etiology | | | |
| HBV | 5 (20.8) | 2 (4.3) | 0.077 ‡ |
| HCV | 7 (29.2) | 20 (43.5) | |
| NBNC | 12 (50.0) | 24 (52.2) | |

**Table 1.** *Cont.*

| Characteristics | PBT (*n* = 24) | TACE + RFA (*n* = 46) | *p* Value |
|---|---|---|---|
| AFP, ng/mL [#] | 4.3 (2.1–13,099.0) | 8.2 (1.1–1922.9) | 0.242 [*] |
| <10 | 16 (66.7) | 25 (54.3) | 0.444 [†] |
| ≥10 | 8 (33.3) | 21 (45.7) | |
| DCP, mAU/mL [#] | 29 (8–22,694) | 38 (7–50,929) | 0.837 [*] |
| <100 | 16 (66.7) | 33 (71.7) | 0.785 [†] |
| ≥100 | 8 (33.3) | 13 (28.3) | |
| Child–Pugh score | | | |
| 5 | 13 (54.2) | 29 (63.0) | 0.608 [†] |
| ≥6 | 11 (45.8) | 17 (37.0) | |
| mALBI grade | | | |
| 1, 2a | 16 (66.7) | 32 (69.6) | 0.794 [†] |
| 2b, 3 | 8 (33.3) | 14 (30.4) | |
| Tumor size, cm [#] | 2.7 (1.2–9.3) | 2.7 (1.4–7.0) | 0.557 [*] |
| <3 | 12 (50.0) | 29 (63.0) | 0.318 [†] |
| ≥3 | 12 (50.0) | 17 (37.0) | |
| Number of treated lesion(s) | | | |
| 1 | 22 (91.7) | 34 (73.9) | 0.201 [‡] |
| 2 | 2 (8.3) | 11 (23.9) | |
| 3 | 0 (0.0) | 1 (2.2) | |
| Vascular invasion | | | |
| Vp 0,1 | 23 (95.8) | 46 (100.0) | 0.343 [†] |
| Vp 2 | 1 (4.2) | 0 (0.0) | |
| BCLC stage | | | |
| 0 | 9 (37.5) | 4 (8.7) | 0.147 [†] |
| A | 7 (29.2) | 34 (73.9) | |
| B | 5 (20.8) | 7 (15.2) | |
| C | 3 (12.5) | 1 (2.2) | |
| Post treatment to target lesion(s) | | | |
| No | 21 (87.5) | 37 (80.4) | 0.526 [†] |
| Yes | 3 (12.5) | 9 (19.6) | |
| TACE | 1 (4.2) | 1 (2.2) | |
| TACE, RFA | 0 (0.0) | 3 (6.5) | |
| TACE, RFA, TKI, HAIC | 0 (0.0) | 1 (2.2) | |
| TACE, TKI or Atezo + Bev | 2 (8.3) | 1 (2.2) | |
| TACE, HAIC | 0 (0.0) | 1 (2.2) | |
| TKI | 0 (0.0) | 1 (2.2) | |
| TKI, HAIC | 0 (0.0) | 1 (2.2) | |
| Post treatment to non-target lesion(s) | | | |
| No | 17 (70.8) | 19 (41.3) | 0.025 [†] |
| Yes | 7 (29.2) | 27 (58.7) | |
| TACE | 3 (12.5) | 3 (6.5) | |
| TACE, TKI | 0 (0.0) | 0 (0.0) | |
| TACE, TKI, HAIC | 1 (4.2) | 1 (2.2) | |
| TACE, RFA (PEIT) | 0 (0.0) | 7 (15.2) | |
| RFA (PEIT) | 1 (4.2) | 15 (32.6) | |
| PBT | 1 (4.2) | 0 (0.0) | |
| Atezo + Bev | 1 (4.2) | 1 (2.2) | |
| Follow-up time, months) | | | |
| Median (range) | 34.9 (4.6–92.5) | 38.8 (3.7–110.8) | 0.329 [*] |

AFP, a-fetoprotein; Atezo + Bev, atezolizumab plus bevacizumab; BCLC stage, Barcelona Clinic Liver Cancer stage; DCP, des-gamma-carboxy prothrombin; ECOG, Eastern Cooperative Oncology Group; HAIC, hepatic arterial infusion chemotherapy; mALBI, modified ALBI; NBNC, nonB-nonC; PBT, proton beam therapy; PEIT, percutaneous ethanol injection therapy; PSM, propensity score matching; RFA, radiofrequency ablation; TACE, transarterial chemoembolization; TKI, tyrosine kinase inhibitor. [#] Continuous variables presented as median (range). [*] Mann–Whitney U test. [†] Fisher's exact test. [‡] Chi-square test.

## 2.2. Etiology of Liver Diseases

The HCC etiology of patients who were positive for anti-hepatitis C virus antibody (HCV Ab) was determined as HCV and the patients who were positive for hepatitis B virus (HBV) surface antigen (HBsAg) was determined as HBV. Patients who were negative for both anti-HCV Ab and HBsAg were defined as nonB-nonC (NBNC).

## 2.3. Assessment for Hepatic Reserve Function

ALBI grade was calculated based on serum albumin and total-bilirubin values using the following formula: ALBI-score: (log10 bilirubin (μmol/L) × 0.66) + (albumin (g/L) × −0.085); and ALBI grade was defined by the score of the following: $\leq -2.60$ = Grade 1, $> -2.60$ to $\leq -1.39$ = Grade 2, $> -1.39$ = Grade 3 [16]. ALBI grade 2 was further divided into 2 subgrades (2a and 2b) using a previously reported cut-off value (ALBI score −2.270) and the 4 ALBI grades were named as mALBI grade [17].

## 2.4. Proton Beam Therapy

Details regarding patient setup, planning images, and proton beam irradiation have been previously reported [18,19]. Respiratory-synchronized 4D-CT (Aquilion LB TSX-201A: Canon Medical Systems Co., Otawara, Japan) was used for planning in combination with the breathing synchronization method in which the up-down respiratory motion of the abdominal skin surface was monitored in real time by the laser sensor, equipped with a respiratory gating system (AZ-733V: Anzai Medical Co., Tokyo, Japan). The CT image for treatment planning was reconstructed at the end of an expiration phase, where the wave signal showed the minimum value, and targets were contoured. Planning target volume (PTV) encompassed the internal target volume with a 0.5 cm margin in all directions. The treatment plan was made using a proton treatment planning system (XiO®-N, Elekta Corp., Stockholm, Sweden) in which the proton dose calculation was performed based on the pencil beam algorithm. The proton dose distribution was formed to the PTV shape based on the passive scattering method. The total dose was set to the geometrical center of the PTV, and two or more beam ports were used for PBT of liver tumors. The proton dose distributions were adjusted to maintain the dose coverage of the PTV as V95% >97.5%, where VD% represents the relative volume receiving at least a specified relative dose, D%. Beam directions were selected to minimize dose to normal liver and to avoid the gastrointestinal (GI) tract by adjusting the field margin for patient's collimator or multi-leaf collimators and the smearing value for patient's bolus. Sometimes the field margin had to be tuned and reduced further when an organ at risk was located near the PTV. In such cases, we aimed to keep the V95 ~100% for the clinical target volume or ITV. The proton beam was delivered based on the treatment planning data using a PBT system (Hi-tachi Corporation, Tokyo, Japan) and the respiratory gating system. Seven protocols were used for respiratory-gated PBT, and the protocols of 66.0–80.5 CGE in 10–38 fractions using an irradiation schedule of 5 fractions per week were employed in this study. The protocol was selected depending on tumor location, based on the previously reported studies [20]. A total dose of 66.0 CGE in 10 fractions was selected for tumors that were not adjacent to the GI tract or the porta hepatis. A total dose of 76.0 CGE in 20 fractions was selected for tumors adjacent to the porta hepatis. For tumors that were adjacent to the GI tract, a total dose of 76.0 CGE in 38 fractions or 70.4 CGE in 32 fractions was selected. Other protocols were employed as needed to minimize the dose for organs at risk or to accommodate the physical condition of the patient.

## 2.5. Transcatheter Arterial Chemoembolization (TACE)

Percutaneous femoral artery puncture was performed under local anesthesia by the Seldinger method and angiography was performed to detect tumor supplying vessels. Conventional TACE (cTACE) was performed by selective catheter insertion into the tumor supplying vessel and by injecting an emulsion containing 30–50 mg of miriplatin (MIRIPLA®; Dainippon Sumitomo Pharma, Osaka, Japan) and 3–10 mL of lipiodol (Lipiodol®; Guerbet

Japan, Tokyo, Japan). Under radiographic guidance, the infusion was discontinued when the flow of lipiodol stopped. Gelatin particles (Gelpart®; Nippon Kayaku Co., Ltd., Tokyo, Japan) were injected until adequate embolization was achieved. Drug-eluting beads TACE (DEB-TACE) was performed using drug-eluting beads (DC Beads®; Eisai Co., Ltd., Tokyo, Japan) that were 100–300 μm in diameter, filled with epirubicin. Microcatheters were inserted into tumor-supplying vessels under a guide wire for embolization and the injection of drug-eluting beads was performed until the tumor staining disappeared. If vascular lakes were detected in the tumor, Gelpart was injected additionally.

### 2.6. Percutaneous Radiofrequency Ablation

Following TACE for HCC, RFA was performed on the same lesion after confirming the therapeutic efficacy of TACE and the patient's good general condition. The median interval between TACE and RFA was 50.5 days. Percutaneous RFA was performed with a 17-G cool-tip needle with a 2- or 3-cm exposed tip (Covidien, Mansfield, MA, USA). RFA was performed under ultrasound guidance after local anesthesia. The ultrasound device was an HI VISION Ascendus with a microconvex 1–6 MHz probe (Hitachi Medical Corporation, Tokyo, Japan). During the RFA procedure, the power output was initially set at 40 W for a 2 cm cautery diameter and 60 W for a 3 cm diameter, respectively, and the power was gradually increased and maintained until the impedance reached its maximum value. Electrode selection and tip length were determined based on tumor size and tumor location. All procedures aimed to obtain a minimum of 5 mm ablation margin around the treated lesion. Contrast-enhanced dynamic liver CT was performed after RFA was completed. If the ablation margin was considered inadequate, an additional RFA was performed during the same hospitalization period.

### 2.7. Evaluation of the Outcomes

Local progression free survival (PFS) was defined as the time from the date of PBT initiation or TACE treatment to the date of local disease progression or death from any cause, whichever occurred first. OS was defined as the time from the date of PBT initiation or TACE treatment to the date of last visit or death, regardless of cause of death. Patients who were lost to follow-up were censored at the last known date of survival, and those who were alive were censored at the data cut-off.

### 2.8. Statistical Analyses

GraphPad Prism version 9.4.0 (GraphPad Software, San Diego, CA, USA) was used for the statistical analyses. Univariate analyses for continuous variables were undertaken using the one-way analysis of variance followed by the Tukey–Kramer post-hoc test. For the analysis of categorical variables, the Mann–Whitney U test, Fisher's exact test, chi-squared test, and log-rank tests were performed. A survival analysis was performed using the Kaplan–Meier method. $p$ values less than 0.05 were considered significant.

## 3. Results

### 3.1. Clinical Course of Representative Cases Treated by PBT or TACE + RFA

There are two HCC cases treated with the curative locoregional therapies (Figure 1). The first case (76 y.o., male) complicated by two lesions (33 mm and 8 mm in diameter) was treated with PBT (Figure 1a). The second case (76 y.o., male) with solitary lesion (32 mm in diameter) was treated with TACE + RFA (Figure 1b). In both cases, HCC tumors were curatively controlled by the treatments. We observed a difference in ALBI score after the treatments between two cases. In the first case treated with PBT, the ALBI score remained in grade 1 for 50 months (Figure 1a). In contrast, in the second case treated with TACE + RFA, the score gradually increased for 38 months (Figure 1b). Thus, the hepatic reserve worsened in the second case treated with TACE + RFA; however, it did not change in the first case treated with PBT.

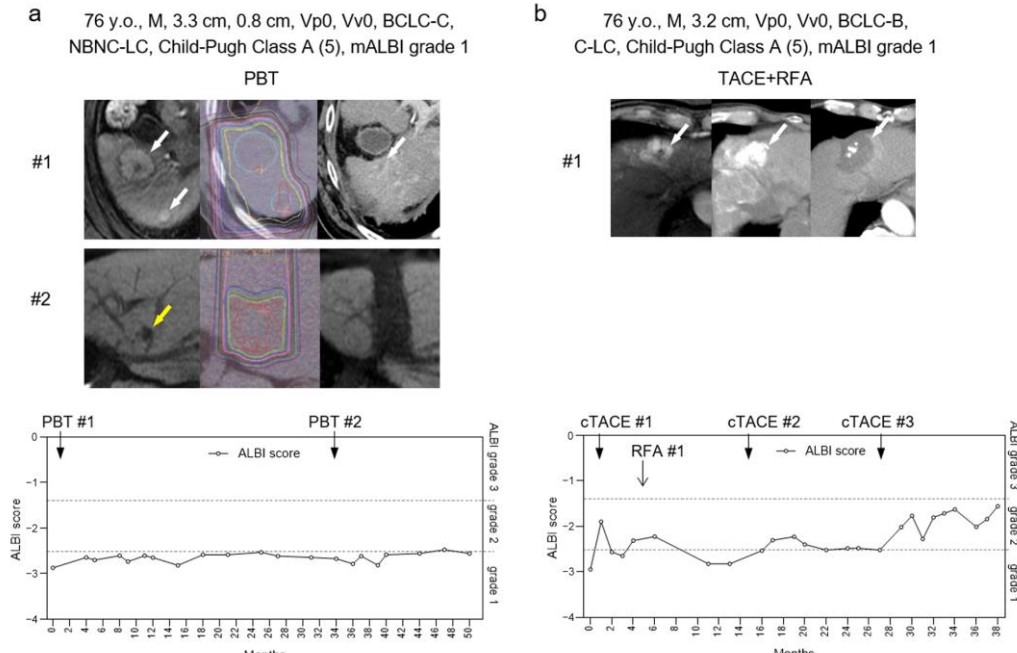

**Figure 1.** Clinical course of two representative cases treated with PBT or TACE + RFA for HCC. (**a**) Imaging studies of one case of PBT for HCC (**#1**; white arrow, **#2**; yellow arrow) (**#1 left**: EOB-MRI arterial phase before treatment, **middle**: PBT dose distribution, **right**: contrast CT scan delayed phase 40 months after PBT, **#2 left**: EOB-MRI hepatocellular phase before treatment, **middle**: PBT dose distribution, **right**: EOB-MRI hepatocellular phase 13 months after PBT), and the course of ALBI score after treatment. (**b**) Imaging studies of one case of TACE + RFA for HCC (white arrow) (**left**: pretreatment EOB-MRI arterial phase, **middle**: simple CT scan after TACE, **right**: contrast CT scan arterial phase after RFA treatment), and the course of ALBI score after treatment.

### 3.2. Therapeutic Efficacy of PBT and TACE + RFA

In the current study, we investigated the course of patients with HCC after treatment with PBT and TACE + RFA. Firstly, we compared the locoregional control ability of PBT and TACE + RFA for patients with HCC. More than 65% of the treated tumor lesions were well-controlled at 60 months in both therapies (Figure 2). Next, we observed that the patients treated with PBT showed 82% OS at 60 months post-treatment. In contrast, the patients treated with TACE + RFA showed 28% OS at 60 months post-treatment. Collectively, we found that PBT demonstrated good locoregional tumor control and longer OS in the patients with HCC that cannot be controlled by hepatectomy or RFA monotherapy compared to TACE + RFA treatment.

### 3.3. Long-Term Changes in Hepatic Reserve after Treatments

The prognosis of patients with HCC is influenced by tumor burden and hepatic reserve [21–23]. Since the levels of tumor markers (AFP and DCP), tumor sizes, numbers and vascular invasion were comparable in patients with HCC treated with PBT and TACE + RFA therapies as seen in Table 1, we evaluated the changes of ALBI score after the treatments in the patients to monitor their hepatic reserve in Figure 3. The ALBI scores in the patients treated with PBT were maintained for 12 months (Figure 3a). In contrast, the scores worsened at 1 month after the treatment by TACE + RFA and the levels did not recover to the baseline ($p < 0.05$). The patients were followed up in our hospital and the additive treatments were performed when tumor recurrence was detected in CT and MRI. We compared the changes of ALBI scores in the patients at the times of the additive second and third treatments (Figure 3b). Similarly, the ALBI scores of PBT-treated patients were maintained at the baseline; however, the scores of TACE + RFA-treated patients worsened with the additive treatments ($p < 0.05$). The results indicated that the hepatic

reserve of patients with HCC was not remarkably influenced by PBT but largely affected by TACE + RFA therapy, leading to the differential prognosis after the two therapies.

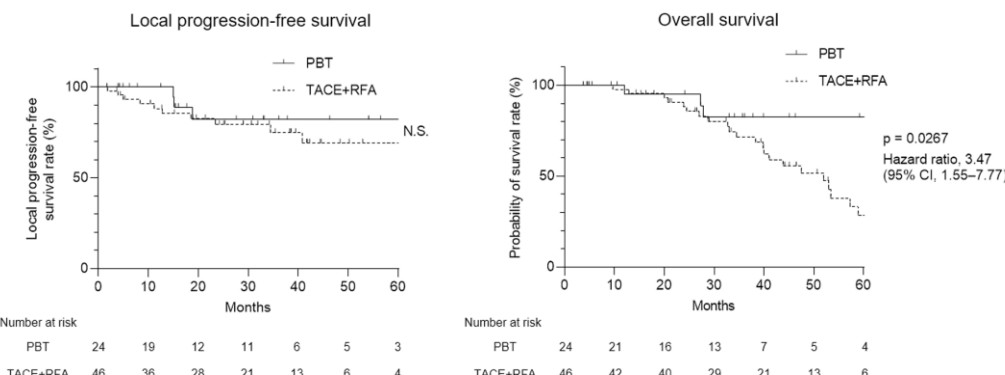

**Figure 2.** Local progression-free survival (PFS) and overall survival (OS) of PBT and TACE + RFA treatment for patients with HCC. (**left**) Kaplan–Meier curves of local PFS and (**right**) OS for PBT and TACE + RFA treatment for patients with HCC. CI, confidence interval; HCC, hepatocellular carcinoma; PBT, proton beam therapy; PFS, progression-free survival; OS, overall survival; TACE, transcatheter arterial chemoembolization; RFA, radiofrequency ablation; N.S., not significant.

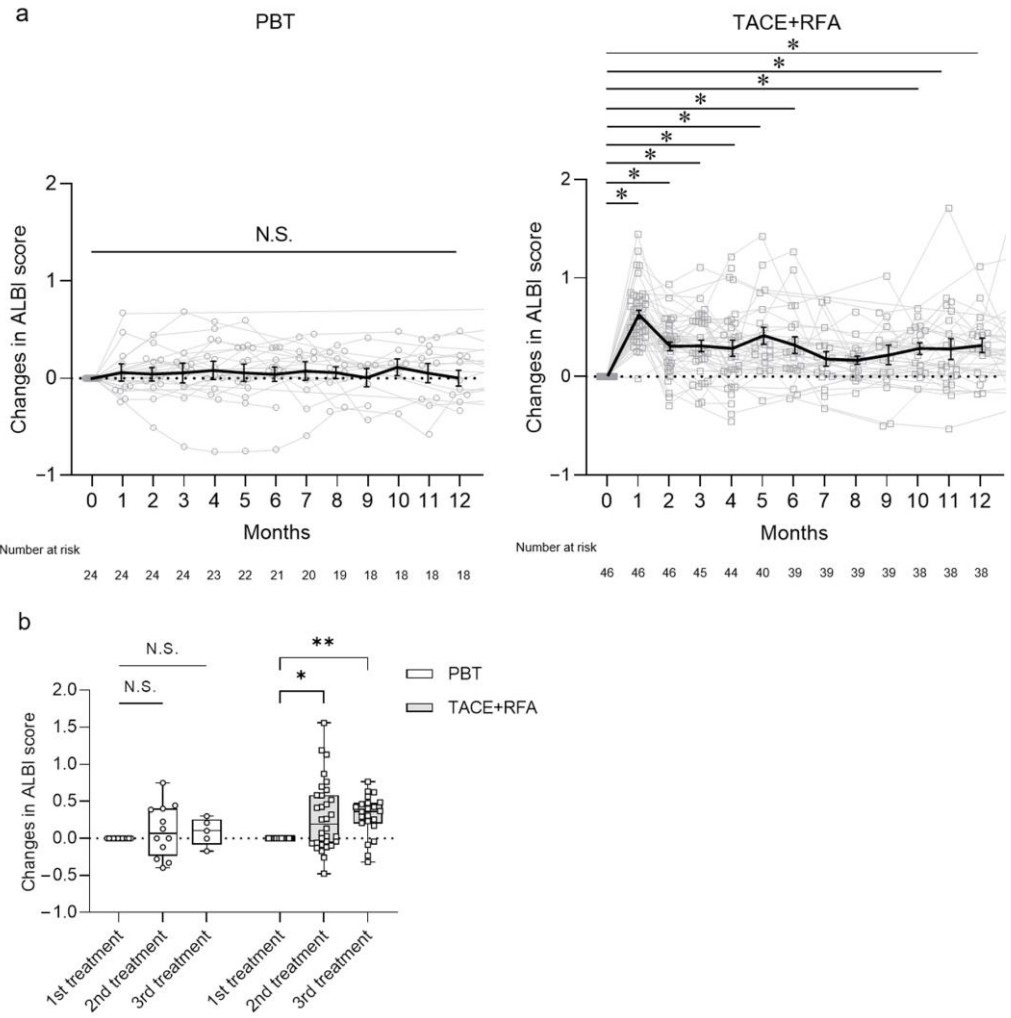

**Figure 3.** Long-term course of ALBI score after PBT and TACE + RFA treatment in patients with HCC. (**a**) Changes in ALBI score after PBT and TACE + RFA treatment for 12 months. Solid black line

represents the means ± SE. (**b**) Changes in ALBI score after PBT and TACE + RFA treatment in second- and third-line treatments of HCC. Each value represents the box and whisker plot (highest, third quartile, median, first quartile and lowest value). (**a**,**b**) Tukey–Kramer post-hoc test. * $p < 0.05$, ** $p < 0.01$, N.S., not significant. ALBI, albumin-bilirubin; HCC, hepatocellular carcinoma; PBT, proton beam therapy; TACE, transcatheter arterial chemoembolization; RFA, radiofrequency ablation.

### 3.4. Clinical Course after Treatments Differentiated by Severity of Hepatic Reserve

We divided the patients into two subgroups based on hepatic reserve, modified ALBI (mALBI) grade, and Child–Pugh score, and monitored their OS after the therapies in Figure 4. The ALBI scores of PBT-treated patients with mALBI grades 1/2a and 2b did not change at the times of the additive second and third treatments (Figure 4a). Although the scores of TACE + RFA-treated patients with mALBI grade 1/2a and 2b did not change at the second treatment, the scores worsened at the third treatment ($p < 0.05$) (Figure 4b). When patients with HCC were treated with PBT, we did not observe a significant difference in OS between the patients with mALBI grades 1/2a and 2b (Figure 4c). In contrast, when treated with TACE + RFA, the patients with mALBI grade 2b showed significantly shorter OS than those with mALBI grades 1/2a (Figure 4d). Similarly, in comparisons between subgroups with Child–Pugh scores of 5 and >6, although there was no significant difference in OS in patients treated with PBT (Figure 4e), we observed significantly shorter OS in patients with Child–Pugh scores of >6 when treated by TACE + RFA (Figure 4f). The data demonstrated that PBT had minimal influence on the hepatic reserve and the prognosis of patients with HCC; however, TACE + RFA had a larger effect on the hepatic reserve and the prognosis of the patients complicated by more severe levels of chronic liver diseases.

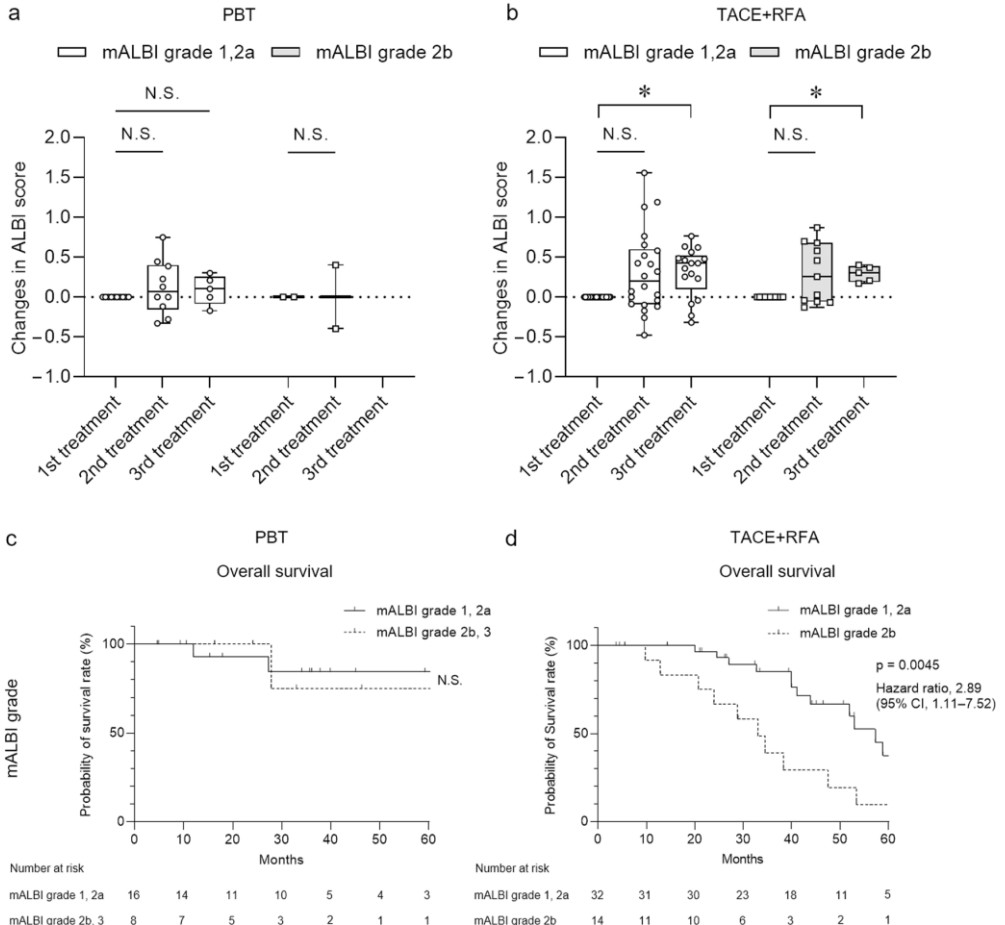

**Figure 4.** *Cont.*

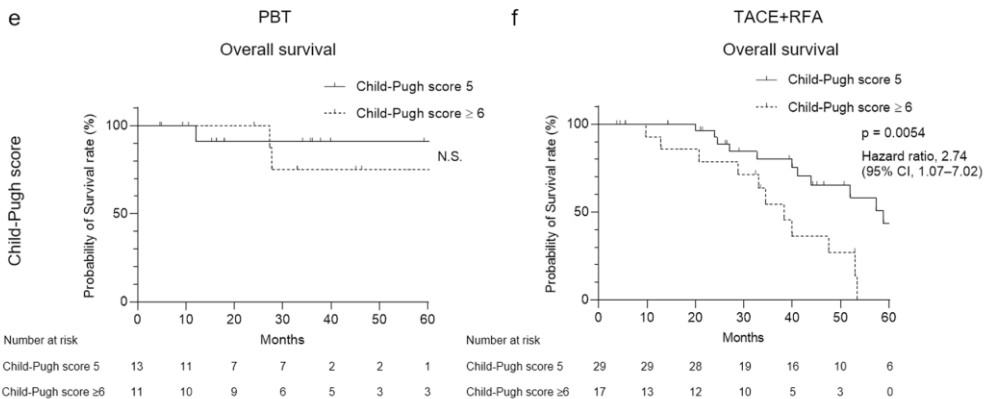

**Figure 4.** Changes in ALBI score and overall survival classified by hepatic reserve after PBT or TACE + RFA treatment. (**a**,**b**) Changes in ALBI score after PBT and TACE + RFA treatment classified by mALBI grade in second- and third-line treatments of HCC. Each value represents the box and whisker plot (highest, third quartile, median, first quartile and lowest value). (**c**–**f**) Overall survival after PBT and TACE + RFA treatment classified by (**c**,**d**) mALBI grade and (**e**,**f**) Child–Pugh score. (**a**,**b**) Tukey–Kramer post-hoc test. * $p < 0.05$, N.S., not significant.

## 4. Discussion

The therapeutic effect of PBT showed good locoregional tumor control and longer OS in patients complicated with HCC that could not be controlled by hepatectomy or RFA monotherapy, compared to TACE + RFA treatment. The hepatic reserve was not influenced by PBT but impaired with TACE + RFA therapy, leading to the difference in prognosis after the two therapies, especially in the patients with more severe levels of chronic liver diseases. These results suggest that PBT may be a beneficial therapeutic tool in patients with HCC and patients with tumors not currently controlled by licensed therapies.

According to the clarifications outlined by the Barcelona Clinic Liver Cancer (BCLC) staging system [24], the patients diagnosed at a very early stage (BCLC 0; single HCC, <2 cm) with a marginal risk of recurrence should be considered for surgical resection; if not, RFA should be the first-line option [25]. The application of curative modalities for HCC such as surgery, transplantation and RFA is often limited by the extent and the location of the tumor, or other patient-related factors. When treating larger lesions (>3 cm), the rates of local tumor recurrence and progression following RFA treatment increase sharply [26], which may be considered for the application of other modalities including TACE.

Tumor factors associated with poor local control of RFA monotherapy were reported as follows: (1) large lesions > 3 cm; (2) adjacent to large vessels such as the portal vein; (3) located at the subphrenic lesion [3,4]. To overcome this problem, the synergistic effects of TACE-derived coagulation necrosis and RFA-induced thermal damage were evaluated for better tumor control and survival outcome in several studies. The combination was developed based on the rationale that TACE can reduce the cooling effect of hepatic blood flow from the hepatic arteries and increase the necrotic effect of RFA therapy [27]. In recent reports, it was shown that TACE combined with RFA (TACE + RFA) therapy could provide significantly better OS, PFS and local tumor control than TACE monotherapy for patients with intermediate-stage (BCLC-B stage) HCC [5,28].

The hepatic reserve was impaired after TACE + RFA therapy in this study. After a single TACE therapy, the deterioration of liver-function-related laboratory values was observed in real-world patients with HCC [29]. In addition to tumor burden, it is well known that the prognosis of patients with HCC complicated by chronic liver diseases is dependent on hepatic reserve function [23,24]. Additionally, a poor hepatic reserve increases the risk of irreversible hepatotoxicity, which may lead to death or the need for urgent liver transplantation [30].

For unresectable large HCC, focal liver radiotherapy (RT) can be used as a treatment option, and technological advancements have facilitated the safe use of highly dose-conformal RT. Since protons have a finite range and generally lead to improved dose distribution, the physical property of PBT can be effectively used to treat large liver tumors in the background of chronic liver disease with reduced hepatic reserve [31,32]. A comparative study of PBT versus intensity-modulated radiation therapy (IMRT) was conducted for the appropriate size criterion for HCC [13]. For macroscopic tumors larger than 6.3 cm in diameter, the average risk of RILD by the Lyman normal-tissue complication probability model was estimated to be 6.2% for PBT vs. 94.5% for IMRT. Consistent with these reports, the present data indicated the advantage of PBT in sparing the non-tumor liver tissues.

Retrospective, non-random design is a major limitation of this research. The therapeutic effect of PBT on unresectable HCC was similar for locoregional tumor control and superior for prolonging overall survival compared with TACE + RFA therapy. Therefore, a prospective randomized controlled trial is necessary to verify these results.

## 5. Conclusions

In summary, PBT may be beneficial for patients with unresectable HCC that could not be controlled by RFA monotherapy, and these patients may show better benefits in terms of PFS and OS than those who received TACE + RFA therapy.

**Author Contributions:** Conceptualization, T.N. (Takuto Nosaka) and Y.N.; validation, T.N. (Takuto Nosaka); investigation, T.N. (Takuto Nosaka), H.M., R.S., Y.A., K.T., T.N. (Tatsushi Naito), M.O., K.K., T.T., Y.S., Y.M., H.T. and Y.N.; data curation, T.N. (Takuto Nosaka); writing—original draft preparation, T.N. (Takuto Nosaka) and Y.N.; writing—review and editing, T.N. (Takuto Nosaka) and Y.N.; visualization, T.N. (Takuto Nosaka) and Y.N.; supervision, Y.N.; project administration, Y.N.; funding acquisition, T.N. (Takuto Nosaka) and Y.N. All authors have read and agreed to the published version of the manuscript.

**Funding:** This research was partially supported by AMED under Grant Numbers JP22fk0210077, JP22fk0210104 and JP22fk0210113, and JSPS KAKENHI Grant-in-Aid for Scientific Research Numbers 22K15992.

**Institutional Review Board Statement:** This retrospective study was approved by The Research Ethics Committee of University of Fukui (approval No. 20220071) and Fukui Prefecture Hospital (approval No. 22-21).

**Informed Consent Statement:** For this type of study, formal consent is not required. Pursuant to the provisions of the ethics committee and the ethics guidelines in Japan, written consent was not required in exchange for public disclosure of study information in the case of retrospective and/or observational studies using a material such as the existing documentation. The study information was open for the public consumption at http://research.hosp.u-fukui.ac.jp/wp-content/uploads/2022/08/20220071.pdf (accessed on 9 August 2022).

**Data Availability Statement:** The data presented in this study are available on request from the corresponding author. The data are not publicly available due to privacy and ethical reasons.

**Conflicts of Interest:** The authors declare no conflict of interest.

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
