# Peer review of "Longer Survival and Preserved Liver Function after Proton Beam Therapy for Patients with Unresectable Hepatocellular Carcinoma"

_curroncol, doi:10.3390/curroncol30040296_

Round 1
Reviewer 1 Report (Previous Reviewer 1)
The Authors have successfully addressed the issues I raised.
Reviewer 2 Report (Previous Reviewer 2)
Thank you to the authors for addressing my previous comments. I have no further comments on the manuscript.
This manuscript is a resubmission of an earlier submission. The following is a list of the peer review reports and author responses from that submission.
Round 1
Reviewer 1 Report
Nosaka et al. aimed to compare the therapeutic effect of proton beam therapy (PBT) on unresectable hepatocellular carcinoma (HCC) that could not be controlled by radio-frequency ablation (RFA) monotherapy was compared with that of transarterial chemoembolization (TACE)+RFA therapy. Their results suggest that, for these patients, PBT is possibly more beneficial than TACE-RFA by sparing more non-tumor liver tissue and therefore liver function. I have several concerns regarding this paper:
1) The reason mentioned in intro for having (presumably) uncontrolled disease by RFA monotherapy is tumor size above 3 cm. However, the majority of their patients (29/46) underwent TACE+RFA despite a tumor size <3 cm, which is suspicious of overtreatment;
2) The Authors claim an impressive 80% overall survival at 84 months with PBT vs. 22% with TACE-RFA group: however, the number at risk table under the Kaplan-Meier plot indicates that this is based on actual follow-up of a single patient who underwent PBT (in fact, only two had a follow-up time >60 months). What were the median and minimum follow-up times in the two groups?
3) No demonstration is given that the small sample size they come out after propensity score matching (51 patients, 19 in the PBT group and 32 in the TACE-RFA group) makes the study sufficiently powered to fulfill their aim.
Reviewer 2 Report
This is a nice study and important to inform the management of patients with a challenging disease. My main comment is that some things are not well defined or explained in the manuscript, so it requires some clarification before publishing.
- Comments on the proton therapy methods: were patients treated with passive scatter or pencil-beam scanning? What RBE value was used to determine CGE? It’s uncommon to use a PTV for proton planning, why was one used in this case? It’s also uncommon to see proton coverage defined to 95% of a target rather than 100%, where does the 5% undercoverage occur – on the distal or proximal or lateral edges? Or was it specifically to spare a nearby critical organ? Was a distal margin applied for range uncertainty?
- Comments relating to Table 1: under the age, there’s a typo, 365 should be >=65. PSM is mentioned here for the first time and not defined until the Results section 3.2; it should be explained in the Methods. HBV and HCV are not defined, and NBNC is defined as “nonB-nonC” which doesn’t make sense if you don’t know what B and C are. Modified ALBI is used in this table but not defined in the Methods; since this is not the same as standard ALBI, it should be defined and references given to justify its use. In this table, Child-Pugh status is broken down by A and B, but in Figure 4, results are plotted for CP = 5 and CP >=6. Since that categorization is used for the results, it should also be used for the table. Table 1 should also include information about follow-up time for the cohorts.
- Question on TACE+RFA: were patients with tumor size < 3cc also treated with both TACE+RFA? The introduction indicates that tumors < 3cc can be treated with RFA alone.